# Lithological Investigation of The Drill Core from a Sedimentary Cover in the Area of the Siljan Ring, Central Sweden

**Vladimir Kutcherov** [1,*], **Olga Sivalneva** [2], **Alexandr Buzilov** [3] **and Alexandr Postnikov** [2]

[1] Department of Energy Technology, KTH Royal Institute of Technology, SE-100 44 Stockholm, Sweden
[2] Department of Lithology, Gubkin Russian State University of Oil and Gas (National Research University), 119991 Moscow, Russia; sivalneva.o@gubkin.ru (O.S.); apostnikov@gubkin.ru (A.P.)
[3] PetroTrace LLC, 115114 Moscow, Russia; alexander.buzilov@ptgeos.com
[*] Correspondence: vladimir.kutcherov@indek.kth.se

**Abstract:** The ring structure of Siljan, located in the central part of Sweden, is considered by many researchers to be a meteorite (impact) crater. Impact craters are among the most complex geological objects on the Earth. The origin and formation of these structures still raises many questions. To find answers to these questions we need reliable geological information about the structure of the crater and the composition of the rocks. Information about the thickness and geological structure of the Siljan Ring area sedimentary cover will help to understand the process of the Siljan Ring structure's formation as well as other similar geological formations on the Earth. Here, we present the results of laboratory studies of sedimentary rock samples taken from four exploration wells drilled in the vicinity of the Siljan Ring crater, which made it possible to compile their detailed lithological description. The laboratory studies included a structural analysis of the samples, and a texture and mineralogical analysis in thin sections. A structure analysis was carried out visually, while structural and mineralogical analyses were carried out on thin sections using a polarizing microscope and a scanning electron microscope. The main components of the rocks (minerals and fragments), along with their ratio and secondary transformations, were determined. The results of the structural analysis of the samples, and the textural and mineralogical analysis of the rocks in thin sections, showed that the sediments' composition in the sedimentary cover near the Siljan Ring structure changes in different areas in accordance with their facies and stratigraphic characteristics. Furthermore, a change in the thickness of the sections and the succession sequences of rock units was established. A change of this nature is presumably caused by tectonic disturbances of an endogenous or impact source.

**Keywords:** Siljan Ring; sedimentary cover; core investigation; lithological types of rock; thin section analysis; impact crater; Paleozoic clastic sediments





## 1. Introduction

A significant number of ring structures have been identified and studied on the Earth's surface. Many of these structures, including the Siljan Ring structure, are considered to be meteorite (impact) craters [1,2]. Impact craters may have huge petroleum potential [3–5]. Giant hydrocarbon deposits associated with these structures have been found throughout the world [6]. Impact craters are among the most complex geological objects on the Earth. The origin and formation of these structures still raises many questions. To find answers to these questions we need reliable geological information about the structure of the crater and the composition of the rocks. The main goal of our study, the results of which are presented in this paper, was to obtain information about the thickness and geological structure of the Siljan Ring area sedimentary cover. This information will help us to understand the process of the Siljan Ring structure's formation as well as other similar geological formations on the Earth.

The Siljan Ring structure has a long history of investigations. A detailed overview of the main results of these investigations can be found in [7]. Detailed geophysical studies [8–12] show the geological complexity of the Siljan Ring structure, i.e., its structural and stratigraphic interrelations. At the same time, a detailed lithological description of the sedimentary succession, the main components of the rocks and their secondary transformations in the crater area are still unresolved.

Sedimentological and stratigraphic information about the sedimentary cover structure in the western and eastern parts of the Siljan Ring, based on the investigation of two core sections (wells VM 1 and Solberg 1), is presented in [13]. Even though the stratigraphic disturbance caused by the two thrust planes complicates the geological interpretation, a classical Ordovician–Silurian carbonate–shale succession was described from the Solberga 1 drill core. This description was used to support a suggestion about the differentiation of Ordovician–Silurian facies belts. It was suggested that the evolution of facies corresponded to the transition from carbonate platform environments to continental conditions [13]. The results of a petrologic study of the core samples from a borehole C-C-1 drilled in the southwestern part of the crater are summarized in [14]. The lithological description allows one to conclude that the sedimentary layers represented by the Upper Ordovician and Silurian sediments have disturbed stratigraphic relations [14].

Here, we present the results of a detailed lithological study of 257 sedimentary rock samples selected from four exploration wells (VM 1, VM 2, Solberga 1, and C-C-1). The research includes textural analysis of samples and the structural and mineralogical analysis of rocks in thin sections. Based on the results obtained, the lithology columns with a description of alteration processes, such as fracturing, and subsequent mineralization were created.

These findings are important for stratigraphy refinement and tectonic setting reconstructions, as well as oil and gas reservoir forecasts.

## 2. Geological Settings

The Siljan Ring structure is located within the Trans-Scandinavian Magmatic Belt of the Baltic Shield. Magmatic rocks are represented mainly by Dala granites (Järna and Siljan types) with sporadically occurring mafic intrusions [15–19]. The crater is a large ring structure with a diameter of about 35 km, well defined in relief. In the center, there is an ancient granite block penetrated by dolerite intrusions (Figure 1). The crater frame is a circular depression occupied by a lake system. The block structure of the crater is described in detail in [20]. The sedimentary cover is preserved only within the depression surrounding the central uplift. The sedimentary cover includes the Paleozoic sequence of Ordovician–Silurian deposits. The thickness of the sedimentary layer was estimated at up to 350 m [21]. In the inner part and outer periphery of the Siljan area, sedimentary strata have opposite dips due to strong tilting and tectonic deformations, which have been considered a consequence of impact [22].

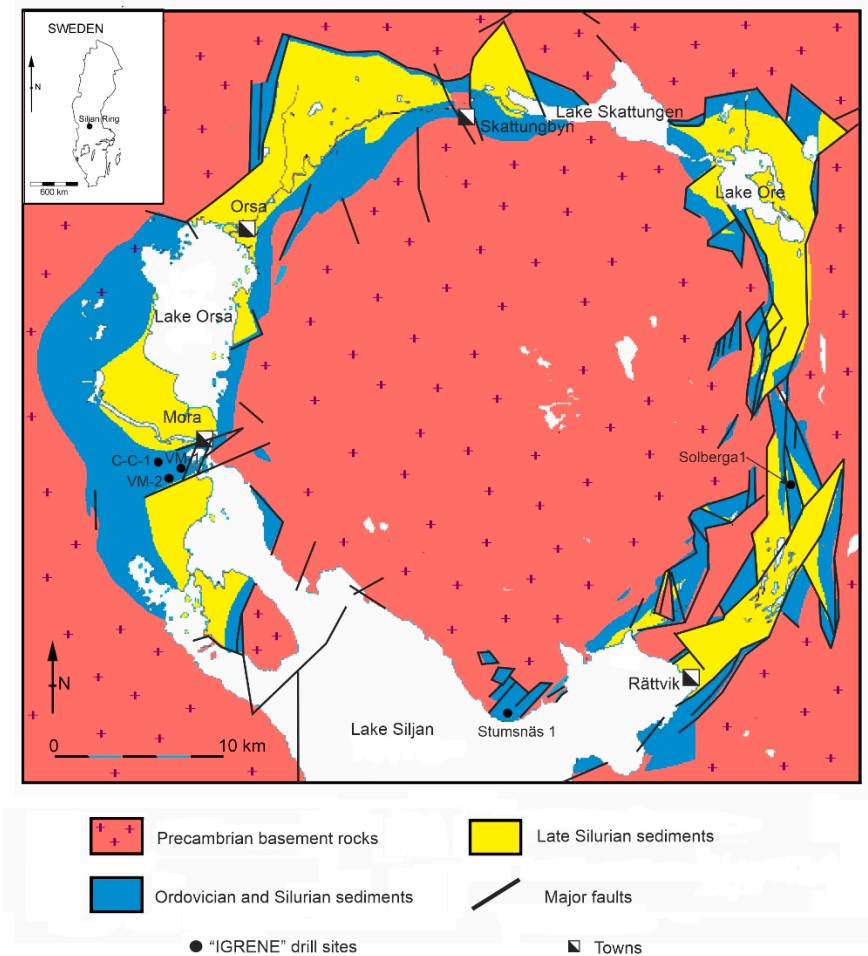

**Figure 1.** Simplified geological map of the Siljan Ring area (modified from Ebbestad and Högström, 2007) [7].

## 3. Materials and Methods

The Swedish company Igrene AB drilled four exploration and three production wells in different parts of the Siljan Ring (Figure 1) as part of a project related to the search for hydrocarbon deposits in the area. The drilling of exploration wells was carried out with a complete selection of core material. The provided core is 1240.58 m long, and the interval of sedimentary cover ranges from 9.85 to 250.9 m for VM-1, 13.9 to 382.3 m for VM-2, 32.6 to 406.15 m for Solberga-1, and 1.95 to 259.53 m for C-C-1.

Detailed lithological descriptions were performed on 257 rock samples selected from four exploration wells as a result of laboratory investigations, which included a structure analysis of samples and texture and mineralogical analysis in thin sections. A structure analysis was carried out on samples (about 5–10 cm in size) visually, while structural and mineralogical analyses were carried out on thin sections using a polarizing microscope (Axio Imager A2m, Carl Zeiss MicroImaging GmbH, München, Germany), as well as a scanning electron microscope (JEOL 6610, Tokyo, Japan). As a result, the main components of the rocks (minerals and fragments), along with their ratio and secondary transformations, were determined.

## 4. Results

Based on the lithological parameters, the rocks were grouped into relatively homogeneous lithological units, and a well column that characterizes the structure of the section was obtained for each unit.

In general, sedimentary cover is represented via Paleozoic sediments and is composed of various types of rocks, namely limestones, sandstones, siltstones, and mudstones. A total

of 19 lithotypes with varying frequencies of occurrence in each section were characterized with samples from four wells (Table 1).

**Table 1.** Rock types' occurrence in the well sections.

| Well Number | VM-1 | VM-2 | C-C-1 | Solberga 1 |
|---|---|---|---|---|
| Sample depth, m | 9.85–250.90 | 13.90–382.30 | 32.60–406.15 | 1.95–259.53 |
| Quantity of m/samples | 241.05<br>42 | 368.40<br>30 | 373.55<br>54 | 257.58<br>131 |
| Rock types | | | | |
| Algal boundstones | ● | | | ● |
| Bryozoan-algal boundstones | | ● | | |
| Wackestones with clotted microfabric | | | | ● |
| Nodular wackestones | ● | ● | | ● |
| Wacke-packstones | ● | ● | ● | ● |
| Calcareous-clayey mudstones | ● | ● | ● | ● |
| Fine-grained limestones | | | ● | |
| Very fine-grained limestones | | | ● | ● |
| Carbonate-clay and carbonate gravelstones | | | ● | ● |
| Clay-carbonate breccias | | | ● | |
| Gravelstones | | | | ● |
| Fine-grained sandstones with pore-filling carbonate cement | | | ● | |
| Very fine-grained sandstones with spotty and complete carbonate cementation | ● | | ● | |
| Very fine-grained carbonate-clayey sandstones | ● | | | |
| Siltstones | ● | | | |
| Calcareous organic-rich mudstones | | | | ● |
| Silty black shales | | | ● | |
| Calcareous shales | ● | | ● | ● |
| Shales | ● | ● | ● | |

Rocks of the same lithotype in different sections are characterized by minor differences in texture and composition, which is primarily due to the zonation changes in facies. For example, in algal boundstones from the Solberga-1 well, algal remains are larger and better preserved than in a similar lithotype from the VM-1 well, and the rocks from the VM-2 well boundstones are represented by the bryozoan-algal type (Figures 2 and 3). Bryozoan-algal bioherms are described for the Freberga Limestone Formation of the early Late Ordovician [23]. In the VM-2 well section, these rocks form a large interval (more than 200 m).

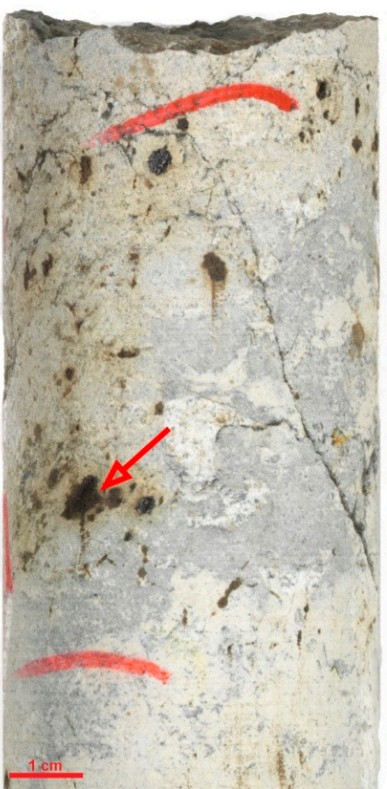

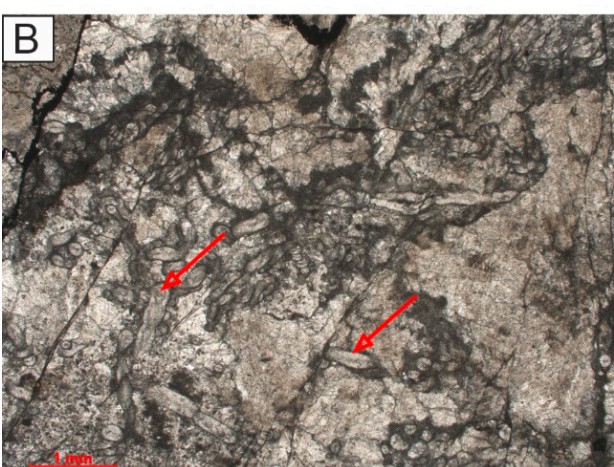

**Figure 2.** Boundstone structure and texture: (**A**) brown spots (red arrow) of oil saturation enclosed in the framework porosity (sample photo); (**B**) large algal plates (red arrows) between calcite cementation zones (thin section photo). Solberga 1 well, depth 105.10 m.

Apparently, at this time, there were favorable shallow marine conditions for the active development of bryozoan-algal bioherms. The Solberga 1 well area is characterized as an area of development for biogenic mound-like structures during the time of the Boda Limestone Formation and towards the end of the Late Ordovician [23–25]. The facies zone in this area may have possessed better conditions for the development of algal bioherm builders, as noted in the more pronounced textural parameters of the algal boundstones.

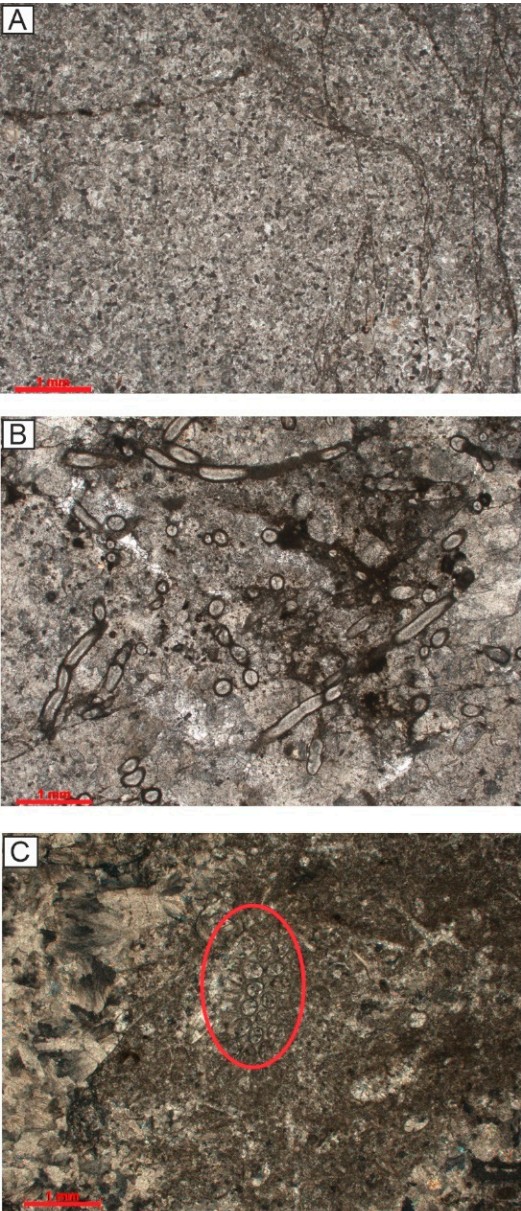

**Figure 3.** Boundstone texture: (**A**) tiny recrystallized algal remains (VM-1 well, sample depth 123.60 m); (**B**) large algal plates (Solberga 1 well, depth 106.80 m); (**C**) bryozoan remains (red circle) in boundstone (VM-2 well, sample depth 248.95 m). Thin section photos. Photo (**C**) is in polarized light.

Sandstones and siltstones are also similar in terms of structural and textural characteristics and mineral composition.

The mineral composition of the clastic part in these rocks is dominated by grains of quartz (85–95%) and feldspars (3–5%). The sandstones are predominantly very fine-grained and have good sorting (Figure 4). There are only minor variations in the grain size and composition of the clasts in different sections: coarser-grained sandstones with more carbonate clasts are distinguished in the section of the C-C-1 well (Figure 5).

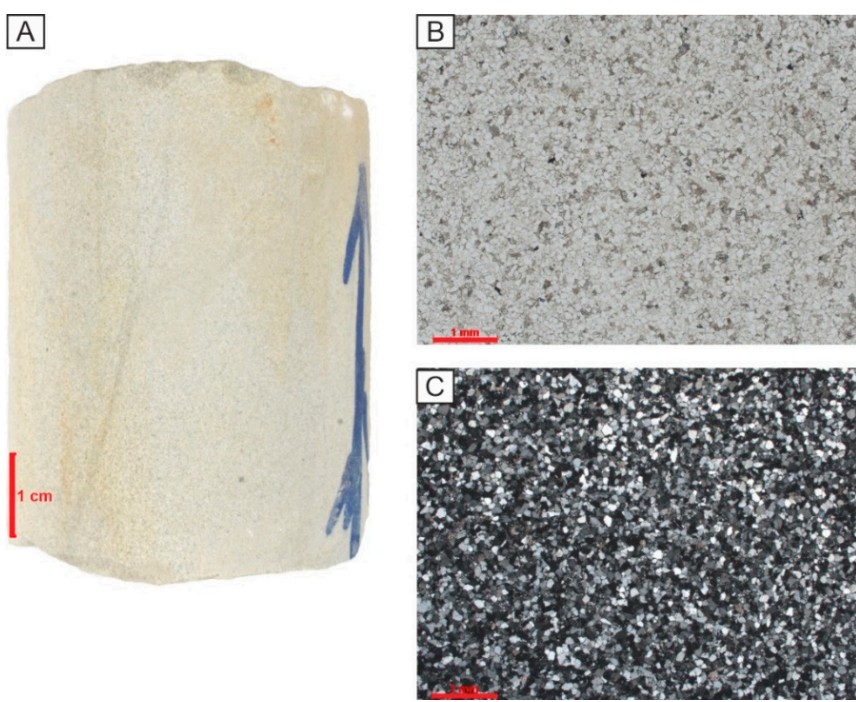

**Figure 4.** Very fine-grained sandstones: (**A**) light color and massive structure of the quartz sandstone; (**B**) good sorting; (**C**) predominantly quartz grains. (**B**,**C**) Thin section photos; photo (**C**) is in polarized light.

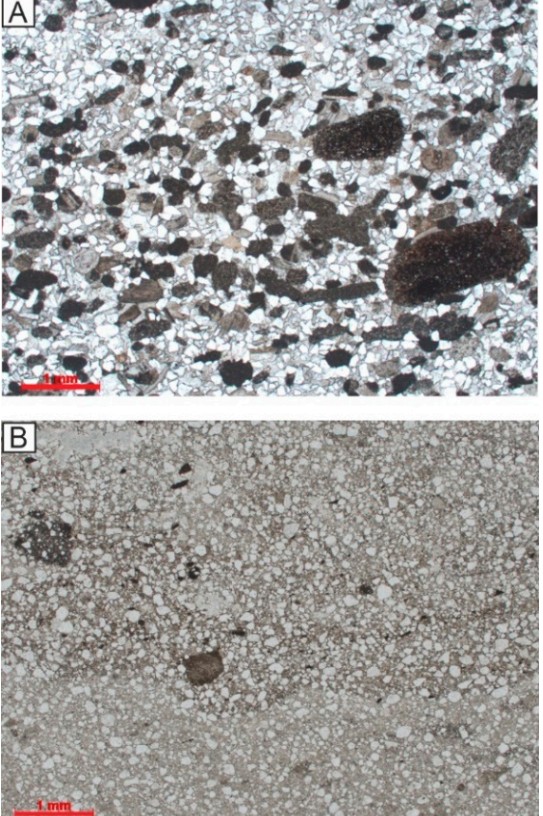

**Figure 5.** Very fine- to fine-grained sandstones: (**A**) large carbonate debris (C-C-1 well, sample depth 34.16 m); (**B**) singular carbonate debris (VM-1 m, sample depth 81.90 m). Thin section photos.

In the sections of the Solberga-1 and C-C-1 wells, interlayers of calcareous organic-rich mudstones are distinguished, the structures of which contain fine organic matter (Figure 6). A comparison of the lithological parameters of these rocks and the rocks of the Fjäcka Shale Formation, identified in the Siljan region [24–26], suggests that they belong to the described deposits. Bituminous inclusions were also found in the cracks of individual samples (Figure 6).

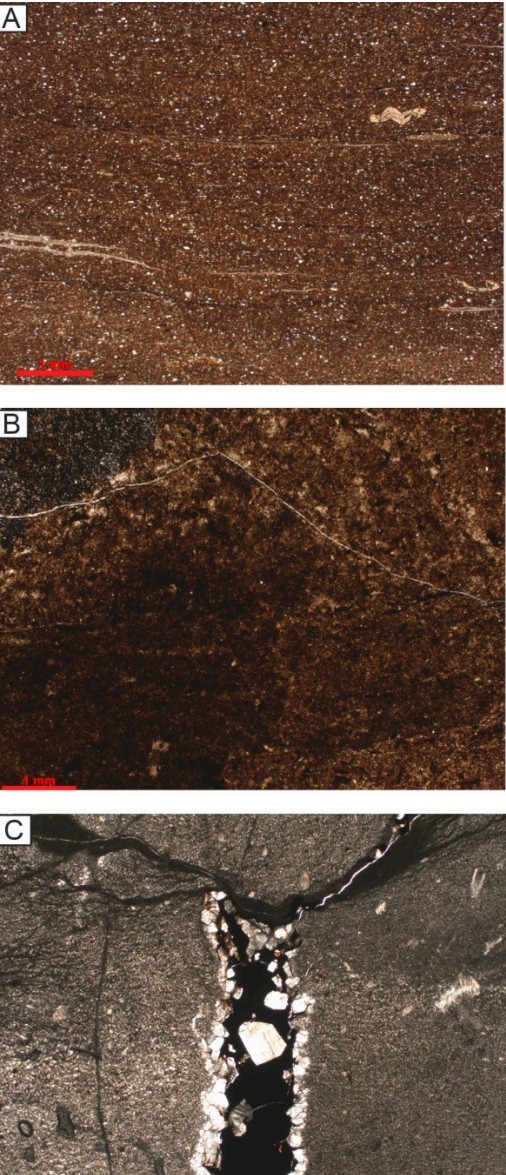

**Figure 6.** Silty black shales and calcareous organic-rich mudstones: (**A**) fine organic matter and shell remains (C-C-1 well, sample depth 329.22 m); (**B**) organic matter clot (Solberga 1 well, sample depth 54.90 m); (**C**) fracture with calcite and bitumen filling (Solberga 1 well, sample depth 257.50 m). Thin section photos.

The section of the C-C-1 well is exceedingly clayey. The calcareous-clayey mudstones, calcareous shales, and shales represent most of the rocks (Figures 7–9). The fine-laminated structures and textures, and small-sized trace fossils, are indicative features of the deepening sublittoral zone, which is not sufficiently deep to be unsuitable for endobenthic organisms.

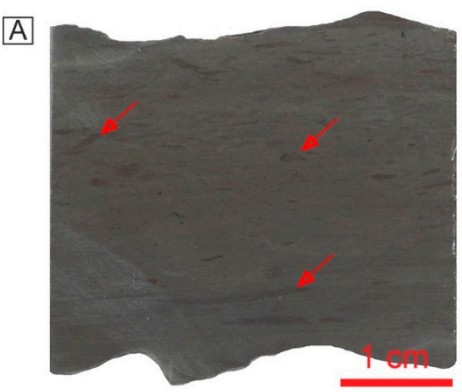

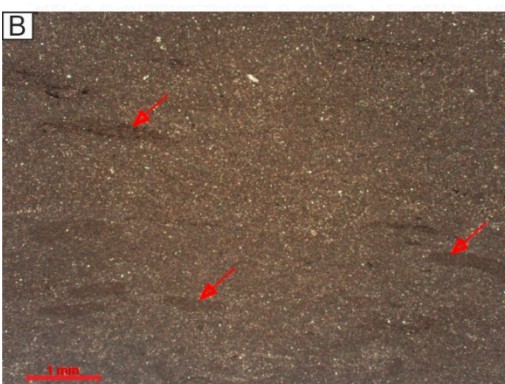

**Figure 7.** Trace fossils (red arrows) in the calcareous-clayey mudstones. (**A**) Sample photo; (**B**) thin section photo. C-C-1 well, sample depth 76.45 m.

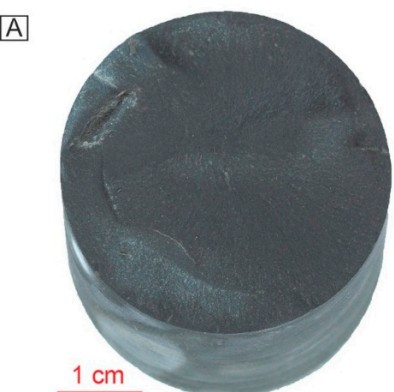

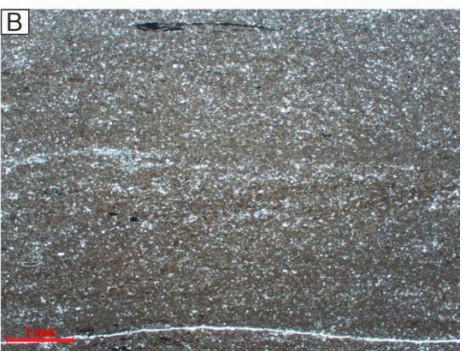

**Figure 8.** Tight calcareous shale with silt admixture. (**A**) Sample photo; (**B**) thin section photo. C-C-1 well, sample depth 356.08 m.

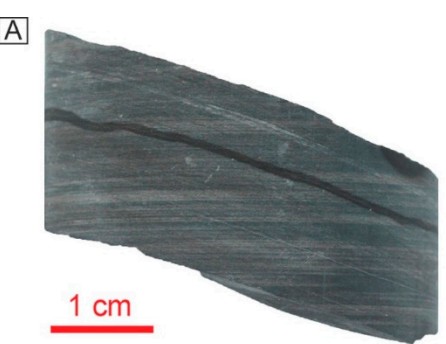

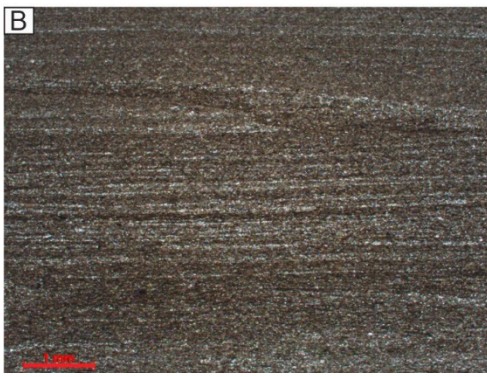

**Figure 9.** Fine-laminated structure and texture in the shale. (**A**) Sample photo; (**B**) thin section photo. C-C-1 well, sample depth 252.39 m.

These types of rocks have a significance presence in the other sections as well, which is why it is possible to speculate about more or less equivalent sedimentary conditions in the different parts of this area.

In almost all rocks, intense secondary processes are observed, among which cracking predominates. The cracks are multidirectional, attenuating, and predominantly filled with clay-carbonate material. In many cases, zones of cataclasis and layer displacement are observed along the cracks (Figures 10 and 11).

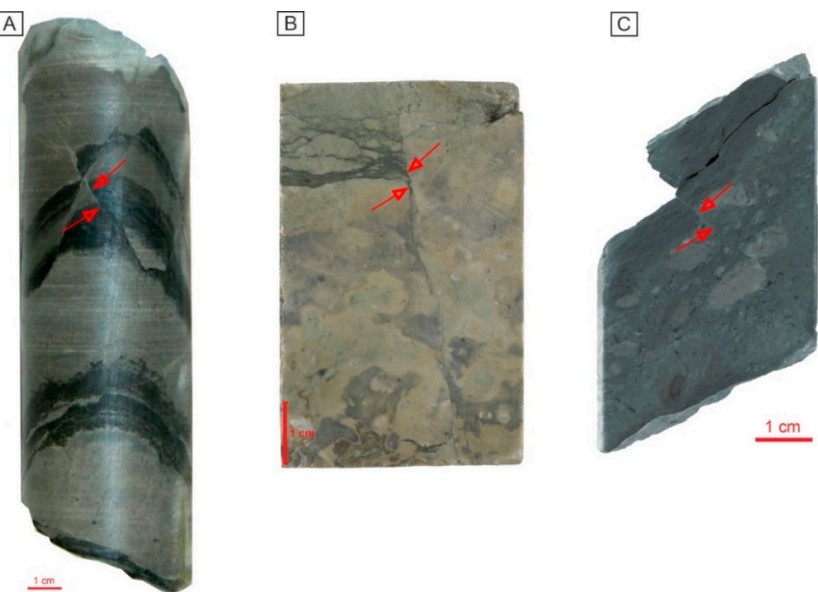

**Figure 10.** Fractures and displacement (red arrows) in the rock samples. (**A**) VM-1 well, (**B**) sample depth 141.25 m; Solberga 1 well, sample depth 160.45 m; (**C**) C-C-1 well, sample depth 396.66 m.

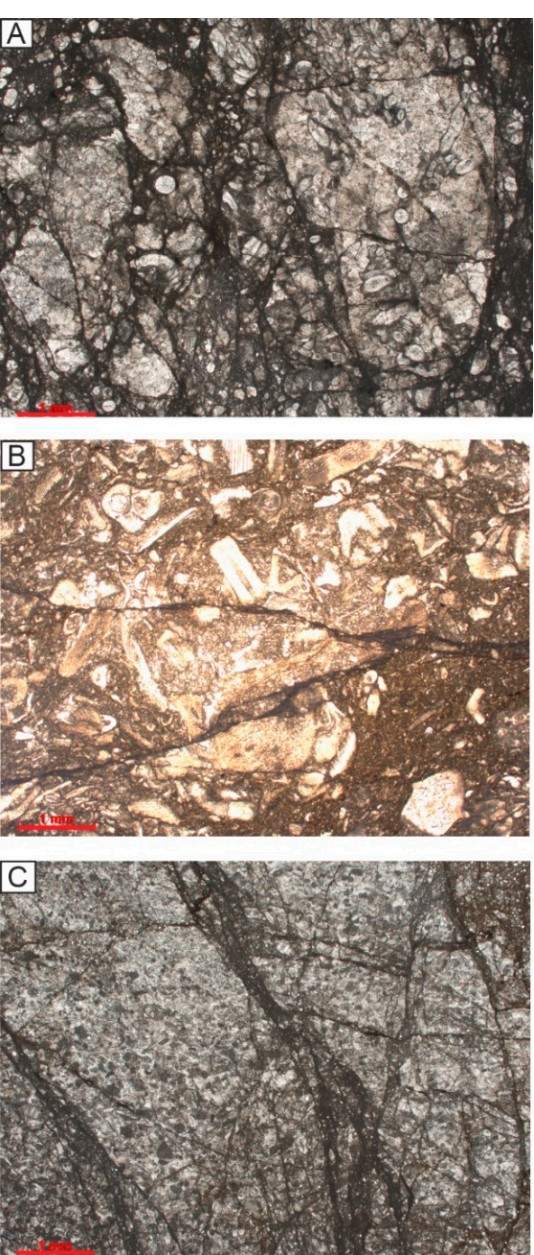

**Figure 11.** Multidirectional fractures: (**A**,**C**) with cataclasis zones (Solberga 1 well, sample depth 114.75 m, VM-1 well, sample depth 119.45 m); (**B**) displacement along the fracture system (C-C-1 well, sample depth 387.93 m). Thin section photos.

## 5. Discussion

In the structure of the sections, differences are observed in both the set of lithotypes and the sequence of their replacement and the overall thicknesses of the identified units (Figure 12, Tables 2 and 3). The thickness of sedimentary cover in the sections of the C-C-1 and VM-2 wells is approximately 100 m higher (368.40 and 373.55 m) than in the sections of the VM-1 and Solberga 1 wells (241.05 and 257.58 m). These differences are due not only to the facies characteristics of the sections, but also due to the manifestation of the tectonic disturbances.

According to the stratigraphic schemes accepted for this territory [7,13,24,25], as well as the stratigraphic description of the VM-1 and Solberga 1 well sections [25], the sandstones of unit 2 of the VM-1 well are Lower Silurian deposits and unconformably cover the underlying limestones. A significant part of the sediments from the Middle Ordovician to the Silurian sections is absent in the section with the VM-1 well [25]. The calcareous

organic-rich mudstones of unit 2 in the Solberga 1 well probably belong to the Fjäcka Shale Formation and occur in a normal sequence with a minor previous break. However, units of similar silty black shales in the lower part of the section of the C-C-1 well occur in a disturbed sequence among the wackestone interlayers. Apparently, they represent allochthonous layers, displaced due to tectonic disturbances. This assumption is confirmed by sharp contacts in the layers with the underlying and overlying limestones, as well as in the appearance of the interlayers of clay-carbonate tectonic breccias (Figure 13).

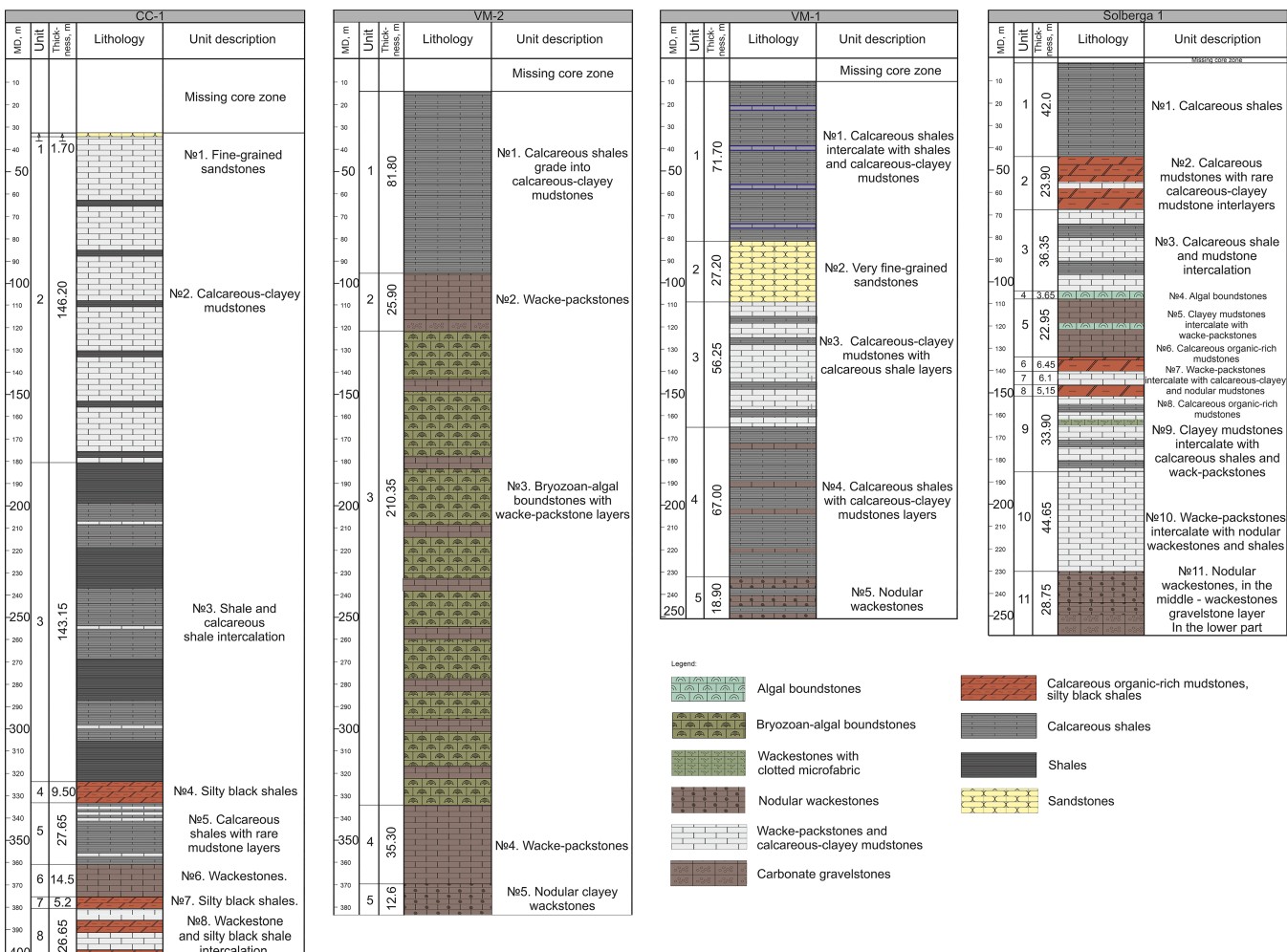

**Figure 12.** Sedimentary cover interval in sections of the C–C–1, VM–2, VM–1, and Solberga 1 wells.

**Table 2.** Unit lithological description for wells: VM–1 and VM–2.

| Unit Number | VM–1 | VM–2 |
|---|---|---|
| 1 | Calcareous shales: intercalate with shales and calcareous-clayey mudstones. Colors: gray, dark gray, brownish gray in the upper part. Structures: massive, fine-laminated; slump, contortion. | Calcareous shales: grade in calcareous-clayey mudstones. Colors: gray, greenish gray, dark gray. Structures: spotty, vaguely laminated. |
| 2 | Very fine-grained sandstones with spotty carbonate cementation grade into siltstones and sandstones with carbonate-clayey cementation. Colors: light gray with brown and dark gray layers in the upper part of the unit. Structures: massive, vaguely laminated, spotty. | Wacke-packstones. Colors: gray, brownish gray, reddish gray. Structures: spotty. Lower part of the unit is the core crushing zone. |

**Table 2.** *Cont.*

| Unit Number | VM–1 | VM–2 |
|---|---|---|
| 3 | Calcareous-clayey mudstones with calcareous shale layers. Rare algal boundstone layers in the middle part of the unit.<br>Colors: gray, dark gray, brownish gray layers in the upper part.<br>Structures: massive, vaguely laminated; rare slump and contortion occurs in the middle part of the unit. | Bryozoan-algal boundstones with wacke-packstone layers.<br>Colors: light gray, gray, brownish gray, reddish gray.<br>Structures: biogenic, spotty. |
| 4 | Calcareous shales with calcareous-clayey mudstones layers.<br>Colors: gray, dark gray.<br>Structures: massive, fine-laminated. | Wacke-packstones.<br>Colors: gray, brownish gray, reddish gray.<br>Structures: spotty. |
| 5 | Nodular wackestones with calcareous shale layers grade in wackestones.<br>Colors: brownish gray, gray, grayish yellow, greenish gray.<br>Structures: spotty, vaguely laminated, nodular. | Nodular clayey wackestones.<br>Colors: reddish gray, gray, brownish gray, yellowish gray.<br>Structures: nodular, spotty. |

**Table 3.** Unit lithological description for wells: Solberga-1 and C–C–1.

| Unit Number | Solberga 1 | C–C–1 |
|---|---|---|
| 1 | Calcareous shales.<br>Colors: gray, dark gray.<br>Structures: massive, vaguely laminated. | Very fine and fine-grained sandstones with carbonate cement.<br>Colors: gray, greenish gray, brownish gray.<br>Structures: spotty, laminated. |
| 2 | Calcareous mudstones with rare calcareous-clayey mudstone interlayers.<br>Colors: brown, dark gray.<br>Structures: massive, rare, spotty (light inclusions), vaguely laminated. | Calcareous-clayey mudstones intercalate with shale layers.<br>Colors: gray, dark gray, brownish gray in the upper part of the unit.<br>Structures: spotty, fine-laminated, nodular, trace fossils. |
| 3 | Calcareous shale and mudstone intercalation; rare interlayers of the wackestones with clotted microfabric in the down part.<br>Colors: gray, dark gray, grayish brown.<br>Structures: massive, vaguely laminated. | Shale and calcareous shale intercalation. Rare clayey mudstone layers.<br>Colors: gray, dark gray.<br>Structures: spotty, fine-laminated, wavy lamination, trace fossils; slump and contortion. |
| 4 | Algal boundstones.<br>Colors: gray, light brown.<br>Structure: inhomogeneous. | Silty black shales.<br>Colors: dark brown, almost black.<br>Structures: fine-laminated, massive. |
| 5 | Clayey mudstones intercalate with wacke-packstones, wackestones with clotted microfabric, and nodular wackestones. Single layer of algal boundstones.<br>Colors: gray, dark gray, grayish brown.<br>Structures: inhomogeneous, spotty. | Calcareous shales with rare mudstone layers and concretions. In the upper part, their appearance increases.<br>Colors: dark gray, gray.<br>Structures: spotty, fine-laminated, nodular, trace fossils. |
| 6 | Calcareous organic-rich mudstones.<br>Color: brown.<br>Structures: massive, rare, and spotty (light inclusions). | Wackestones.<br>Color: gray, greenish dark gray, brownish gray.<br>Structures: spotty, nodular. |
| 7 | Wacke-packstones intercalate with calcareous-clayey and nodular mudstones.<br>Colors: gray, dark gray.<br>Structures: spotty, nodular. | Silty black shales.<br>Colors: dark brown, almost black.<br>Structures: fine-laminated, massive. |

**Table 3.** *Cont.*

| Unit Number | Solberga 1 | C–C–1 |
|:---:|---|---|
| 8 | Calcareous organic-rich mudstones.<br>Color: brown.<br>Structures: massive, rare, and spotty (light inclusions). | Wackestone and silty black shale intercalation.<br>Colors: wackestones—gray, dark gray; shales—dark brown, almost black.<br>Structures: wackestones—nodular, spotty; shales—fine-laminated. Sometimes slump and contortion structures occur. |
| 9 | Clayey mudstones intercalate with calcareous shales and wacke-packstones. Single layer of the carbonated gravelstones. Middle part consist of wackestones with clotted microfabric.<br>Colors: gray, dark gray.<br>Structures: massive, vaguely laminated, spotty. | |
| 10 | Wacke-packstones intercalate with nodular wackestones and shales.<br>Colors: gray, dark gray.<br>Structures: massive, vaguely laminated, spotty. | |
| 11 | Nodular wackestones, in the middle part—wackestones. In the lower part—gravelstone layer.<br>Colors: gray, brownish gray, dark gray, grayish yellow, grayish brown.<br>Structures: nodular, spotty, vaguely laminated. | |

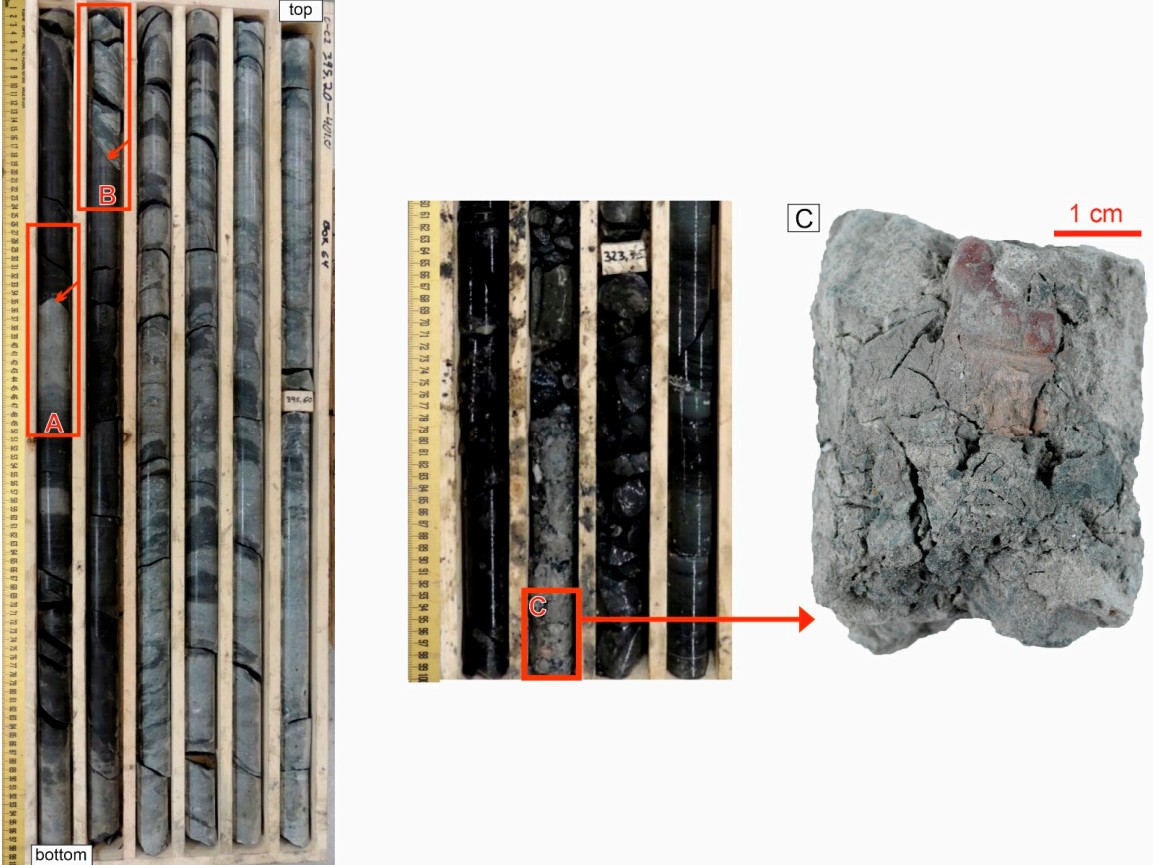

**Figure 13.** (**A**,**B**) Sharp contact of the calcareous organic-rich mudstones and wackestones. (**C**) Clay-carbonate breccia. C-C-1 well. Core photo (interval depth 395.20–401.00 m) and rock sample (sample depth 323.89 m).

The layer of bryozoan-algal boundstones with interlayers of wacke-packstones in the section of the VM-2 well has a significant thickness of 210.35 m. It can be assumed that these limestones belong to the Late Ordovician Freberga Limestone Formation [7,23,24], and this interval is a complex of biohermal structures, which cyclically replace each other in the section.

## 6. Conclusions

The results of this drill core investigation allow us to make the following conclusions:

1.  The composition of sediments in the sedimentary cover on the territory of the Siljan structure changes in different areas in accordance with their facies and stratigraphic characteristics. Along with this, a change in the thickness of the sections and the sequence of the succession of rock units was established. This type of change is presumably due to the tectonic disturbances of an endogenous and impact nature.
2.  The sedimentary sections are not complete as a result of tectonic displacements, and a significant part of the Lower and Middle Ordovician succession is missing. The presence of Fjäcka Shale layers between the Upper Ordovician and Lower Silurian sediments is suggested to be a result of thrust formation, and there is a possibility of determining fault planes.
3.  The structure, textures, and composition of the sedimentary rocks suggest the presence of deposition in the sublittoral zone within a marine basin with unfavorable conditions for high-relief organic buildup development. The presence of the significant amount of such types of rocks, such as the calcareous-clayey mudstones, calcareous shales, and shales, is the evidence to support this conclusion. Hence, black shale and shale layers were likely formed at the time of maximum transgression.
4.  The presence of organic-rich rocks and bituminous inclusions allows us to suggest the possibilities of hydrocarbon formation and migration, which need to be investigated more thoroughly.
5.  The complexity of fracture patterns and secondary processes along these should be the object of further detailed studies in order to reveal the nature and sequence of the tectonic disturbances.

**Author Contributions:** Conceptualization, A.P., O.S., V.K. and A.B.; methodology, A.P., O.S. and A.B.; investigation, O.S. and A.B.; resources, V.K.; writing—original draft preparation, O.S. and V.K.; writing—review and editing, O.S. and V.K.; visualization, O.S. and A.B.; supervision, A.P., O.S. and V.K. All authors have read and agreed to the published version of the manuscript.

**Funding:** This research received no external funding.

**Data Availability Statement:** All original data are present in this paper.

**Acknowledgments:** The authors thank their colleagues from the Department of Lithology (Gubkin University of Oil and Gas) for their helpful comments and suggestions. The authors also appreciate the comments and constructive suggestions of all reviewers.

**Conflicts of Interest:** The authors declare no conflict of interest.

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
