# Peer review of "Lithological Investigation of The Drill Core from a Sedimentary Cover in the Area of the Siljan Ring, Central Sweden"

_geosciences, doi:10.3390/geosciences14010001_

Round 1
Reviewer 1 Report
Comments and Suggestions for Authors
Please see the attached annotated pdf file for detailed comments. Few key suggestions are as follows:
Abstract section
1. enlist and describe the key objectives of this study and include the data and methods statement in the abstract section.
2. vague and unclear results reported in the abstract section. detail the heterogeneity in texture, structure, and lithology of sedimentary rocks.
3. what is the global impact and significance of this report? please include this statement in the abstract section.
Geological settings section
4. Figure 1: change the color scheme and the Regional map is too small. Please redraw the figure.
5. cite the literature used to draw Figure 1 in the figure caption.
6. The data and methods section is missing. please include this section.
Results section
7. Table 1: what is the description of shales given in the last row? it is very unclear if you only mention shale in describing the lithology.
8. Figure 2: label the figure and increase the text to show a clear scale of the image.
9. Figure 6: very small font size and too much use of grey color. it is very unlikely to differentiate all the lithologies. redraw the figure.
10. Include more figures and images of core samples for a detailed description of various lithologies.
References section
11. check the format of the bibliography required for the journal and modify all the references accordingly.
12. simlarity index excluding the bibliography is 29% that is high. it should be significantly reduced in the revised version.

Author Response
Thank you very much for taking the time to review this manuscript. Please find the detailed responses below and the corresponding corrections highlighted changes in the re-submitted files.

Reviewer 2 Report
Comments and Suggestions for Authors
see attached

Author Response
Thank you very much for taking the time to review this manuscript. Please find the detailed responses below and the corresponding corrections highlighted changes in the resubmitted files.

Round 2
Reviewer 1 Report
Comments and Suggestions for Authors
The title of the manuscript should be revised as "Lithological Investigation of The Drill Core from 2 Sedimentary Cover in The Siljan Ring, Central 3 Sweden".
add the keywords: impact crater; Paleozoic clastic sediments.
Figure 13: mark the sharp contact between clastic and carbonate sediments. and label the figure of the core sample with the details.
Author Response
Dear reviewer, thank you very much for taking the time again to review this manuscript. The suggested changes have been made:
1) The title has been revised;
3) The key words have been added;
4) Figure 13 has been corrected.
Please find in the attachment a new revised version of the manuscript.
